# Lookahead Pathology in Monte-Carlo Tree Search

**Primary Keywords:** *(7) Multi-Agent Planning*

## Abstract

Monte-Carlo Tree Search (MCTS) is an adversarial search paradigm that first found prominence with its success in the domain of computer Go. Early theoretical work established the game-theoretic soundness and convergence bounds for Upper Confidence bounds applied to Trees (UCT), the most popular instantiation of MCTS; however, there remain notable gaps in our understanding of how UCT behaves in practice. In this work, we address one such gap by considering the question of whether UCT can exhibit *lookahead pathology* — a paradoxical phenomenon first observed in Minimax search where greater search effort leads to worse decision-making. We introduce a novel family of synthetic games that offer rich modeling possibilities while remaining amenable to mathematical analysis. Our theoretical and experimental results suggest that UCT is indeed susceptible to pathological behavior in a range of games drawn from this family.

## 1 Introduction

Monte-Carlo Tree Search (MCTS) is an online planning framework that first found widespread use in game-playing applications (Coulom 2007; Finnsson and Björnsson 2008; Arneson, Hayward, and Henderson 2010), culminating in the spectacular success of AlphaGo (Silver et al. 2016, 2017). MCTS-based approaches have since been successfully adapted to a broad range of other domains, including combinatorial search and optimization (Sabharwal, Samulowitz, and Reddy 2012; Goffinet and Ramanujan 2016), malware analysis (Sartea and Farinelli 2017), knowledge extraction (Liu et al. 2020), and molecule synthesis (Kajita, Kinjo, and Nishi 2020).

Despite these high-profile successes, however, there are still aspects of the algorithm that remain poorly understood. Early theoretical work by Kocsis and Szepesvári introduced the Upper Confidence bounds applied to Trees (UCT) algorithm, now the most widely used variant of MCTS. Their work established that in the limit, UCT correctly identified the optimal action in sequential decision-making tasks, with the regret associated with choosing sub-optimal actions increasing at a logarithmic rate (Kocsis and Szepesvári 2006). Coquelin and Munos, however, showed that in the worst-case scenario, UCT's convergence could take time super-exponential in the depth of the tree (Coquelin and Munos 2007). More recent work by Shah, Xie, and Xu proposes a "corrected" UCT with better convergence properties (Shah, Xie, and Xu 2020).

In parallel, there has been a line of experimental work that has attempted to understand the reasons for UCT's success in practice and characterize the conditions under which it may fail. Ramanujan, Sabharwal, and Selman considered the impact of *shallow traps* — highly tactical positions in games like Chess that can be established as wins for the opponent with a relatively short proof tree — and argued that UCT tended to misevaluate such positions (Ramanujan, Sabharwal, and Selman 2010a; Ramanujan and Selman 2011). Finnsson and Björnsson considered the performance of UCT in a set of artificial games and pinpointed *optimistic moves*, a notion similar to shallow traps, as a potential Achilles heel (Finnsson and Björnsson 2011). James, Konidaris, and Rosman studied the role of random playouts, a key step in the inner loop of the UCT algorithm, and concluded that the smoothness of the payoffs in the application domain determined the effectiveness of playouts (James, Konidaris, and Rosman 2017). Our work adds to this body of empirical research, but is concerned with a question that has thus far not been investigated in the literature: *can UCT behave pathologically?*

The phenomenon of *lookahead pathology* was first discovered and analyzed in the 1980s in the context of planning in two-player adversarial domains (Beal 1980; Nau 1982; Pearl 1983). Researchers found that in a family of synthetic board-splitting games, deeper Minimax searches counter-intuitively led to worse decision-making. In this paper, we present a novel family of abstract, two-player, perfect information games, inspired by the properties of real games such as Chess, in which UCT-style planning displays lookahead pathology under a wide range of conditions.

## 2 Background

### Monte-Carlo Tree Search

Consider a planning instance where an agent needs to determine the best action to take in a given state. An MCTS algorithm aims to solve this problem by iterating over the following steps to build a search tree.

- **Selection:** Starting from the root node, we descend the tree by choosing an action at each level according to some policy $\pi$. UCT uses UCB1 (Auer, Cesa-Bianchi,

and Fischer 2002), an algorithm that optimally balances exploration and exploitation in the multi-armed bandit problem, as the selection policy. Specifically, at each state $s$, UCT selects the action $a = \pi(s)$ that maximizes the following upper confidence bound:

$$\pi(s) = \operatorname*{argmax}_{a} \left( \overline{Q}(T(s,a)) + c \cdot \sqrt{\frac{\log n(s)}{n(T(s,a))}} \right)$$

Here, $T(s,a)$ is the transition function that returns the state that is reached from taking action $a$ in state $s$, $\overline{Q}(s)$ is the current estimated utility of state $s$, and $n(s)$ is the visit count of state $s$. The constant $c$ is a tunable exploration parameter. In adversarial settings, the negamax transformation is applied to the UCB1 formula, to ensure that utilities are alternatingly maximized and minimized at successive levels of the search tree.

- **Evaluation:** The recursive descent of the search tree using $\pi$ ends when a node $s'$ that is previously unvisited, or that corresponds to a terminal state (i.e., one from which no further actions are possible), is reached. If $s'$ is non-terminal, then an estimate $R$ of its utility is calculated. This calculation may take the form of random playouts (i.e., the average outcome of pseudorandom completions of the game starting from $s'$), handcrafted heuristics, or the prediction of a learned estimator like a neural network. For terminal nodes, the true utility of the state is used as $R$ instead. The node $s'$ is then added to the search tree, so that the size of the search tree grows by one after each iteration.

- **Backpropagation:** Finally, the reward $R$ is used to update the visit counts and the utility estimates of each state $s$ that was encountered on the current iteration as follows:

$$\overline{Q}(s) \leftarrow \frac{n(s)\overline{Q}(s) + R(s)}{n(s) + 1} \qquad n(s) \leftarrow n(s) + 1$$

This update assigns to each state the average reward accumulated from every episode that passed through it.

We repeat the above steps until the designated computational budget is met; at that point, the agent selects the action $a = \operatorname{argmax}_{a'} Q(T(r, a'))$ to execute at the root node $r$.

## Lookahead Pathology

Searching deeper is generally believed to be more beneficial in planning domains. Indeed, advances in hardware that permitted machines to tractably build deeper Minimax search trees for Chess were a key reason behind the success of Deep Blue (Campbell, Hoane, and hsiung Hsu 2002). However, there are settings in which this property is violated, such as the P-games investigated by several researchers in the 1980s (Beal 1980; Nau 1982; Pearl 1983). Over the years, many have attempted to explain the causes of pathology and why it is not encountered in real games like Chess. Nau et al. reconciled these different proposals and offered a unified explanation that focused on three factors: the branching factor of the game, the degree to which the game demonstrates *local similarity* (a measure of the correlation in the utilities of nearby states in the game tree), and the *granularity* of the

heuristic function used (the number of distinct values that the heuristic takes on) (Nau et al. 2010). They concluded that pathology was most pronounced in games with a high branching factor, low local similarity and low heuristic granularity.

## Synthetic Game Tree Models

There is a long tradition of using abstract, artificial games to empirically understand the behavior of search algorithms. The P-game model is a notable example, that constructs a game tree in a bottom-up fashion (Pearl 1980). To create a P-game instance, the values of the leaves of the tree are carefully set to win/loss values (Pearl 1980), though variants using real numbers instead have also been studied (Luštrek, Gams, and Bratko 2005). The properties of the tree arise organically from the distribution used to set the leaf node values. P-games were the subject of much interest in the 1980s, as the phenomenon of lookahead pathology was first discovered in the course of analyzing the behavior of Minimax search in this setting (Beal 1980; Nau 1982). While the model's relative simplicity allows for rigorous mathematical analysis, P-games also suffer from a couple of drawbacks. Firstly, computing the value of the game and the minimax value of the internal nodes requires search, and that all the leaf nodes of the tree be retained in memory, which restricts the size of the games that may be studied in practice. Secondly, the construction procedure only models a narrow class of games, namely, ones where the values of leaf nodes are independent of each other.

Other researchers have proposed top-down models, where each internal node of the tree maintains some state information that is incrementally updated and passed down the tree. The value of a leaf node is then determined by a function of the path that was taken to reach it. For example, in the models studied by Nau and Scheucher and Kaindl, values are assigned to the edges in the game tree and the utility of a leaf node is determined by the sum of the edge values on the path from the root node to the leaf (Nau 1983; Scheucher and Kaindl 1998). These models were used to demonstrate that correlations among sibling nodes were sufficient to eliminate lookahead pathology in Minimax. However, search is still required to determine the true value of internal nodes, thereby only allowing for the study of small games. Furtak and Buro proposed *prefix value trees* that extend the model of Scheucher and Kaindl by observing that the minimax value of nodes along a principal variation can never worsen for the player on move (Furtak and Buro 2009). Setting the values of nodes while obeying this constraint during top-down tree construction obviates the need for search, which allowed them to generate arbitrarily large games.

Finally, synthetic game tree models have also been used to study the behavior of MCTS algorithms like UCT. For example, Finnsson and Björnsson used variations of Chess to identify the features of the search space that informed the success and failure of UCT (Finnsson and Björnsson 2011). Ramanujan, Sabharwal, and Selman studied P-games augmented with "critical moves" — specific actions that an agent must get right at important junctures in the game to ensure victory (Ramanujan, Sabharwal, and Selman 2010b).

We refine this latter idea and incorporate it into a top-down model, which we present in the following section.

## 3 Critical Win-Loss Games

Our goal in this paper is to determine the conditions under which UCT exhibits lookahead pathology. To conduct this study, we seek a class of games that satisfy several properties. Firstly, we desire a model that permits us to construct arbitrarily large games to more thoroughly study the impact of tree depth on UCT's performance. We note that most of the game tree models discussed in Section 2 do not meet this requirement. The one exception is the prefix value tree model of Furtak and Buro that, however, fails a different test: the ability to construct games with parameterizable difficulty. Specifically, we find that prefix value games are too "easy" as evidenced by the fact that a naïve planning algorithm that combines minimal lookahead with purely random playouts achieves perfect decision-making accuracy in this setting (see Appendix A for a proof of this claim). In this section, we describe *critical win-loss games*, a new generative model of extensive-form games, that addresses both these shortcomings of existing models.

### Game Tree Model

Our model generates game trees in a top-down fashion, assigning each node a true utility of either $+1$ or $-1$. In principle, every state in real games like Chess or Go can be labeled in a similar fashion (ignoring the possibility of draws) with their true game-theoretic values. Thus, we do not lose any modeling capacity by limiting ourselves to just win-loss values. The true minimax value of a state imposes constraints on the values of its children as noted by Furtak and Buro — in our setting, this leads to two kinds of internal tree nodes. A *forced node* is one with value $-1$ $(+1)$ at a maximizing (minimizing) level. All the children of such a node are constrained to also be $-1$ $(+1)$. A *choice node*, on the other hand, is one with value $+1$ $(-1)$ at a maximizing (minimizing) level. At least one child of such a node must have the same minimax value as its parent. Figure 1 presents examples of these concepts.

All the variation that is observed in the structures of different game trees is completely determined by what happens at choice nodes. As noted earlier, *exactly* one child of a choice node must share the minimax value of its parent; the values of the remaining children are unconstrained. We introduce a parameter called the *critical rate* ($\gamma$) that determines the values of these unconstrained children. We now describe our procedure for growing a critical win-loss tree rooted at a node $s$ with minimax value $v(s)$:

- Let $S = \{s_1, s_2, \ldots, s_b\}$ denote the $b$ children of $s$.
- If $s$ is a forced node, then we set $v(s_1) = v(s_2) = \ldots = v(s_b) = v(s)$, and continue recursively growing each subtree.
- If $s$ is a choice node, then we pick an $s_i \in \{s_1, \ldots, s_b\}$ uniformly at random and set $v(s_i) = v(s)$, designating $s_i$ to be the child that corresponds to the optimal action choice at $s$. For all $s_j$ such that $j \neq i$, we set $v(s_j) = -v(s)$ with probability $\gamma$ and we set $v(s_j) = v(s)$ with

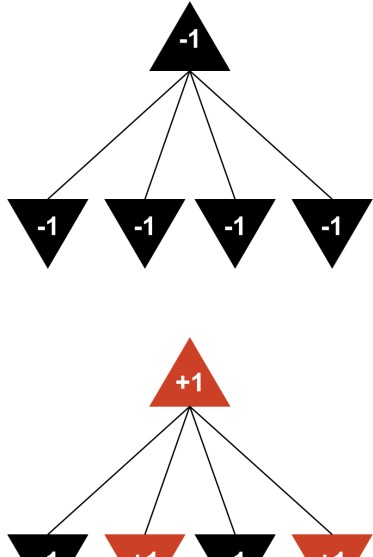

Figure 1: An example of a forced node (left) and a choice node (right). Upward-facing triangles represent maximizing nodes while downward-facing triangles represent minimizing nodes.

probability $1 - \gamma$, before recursively continuing to grow each subtree.

We make several observations about the trees grown by this model. Firstly, one can apply the above growth procedure in a lazy manner, so that only those parts of the game tree that are actually reached by the search algorithm need to be explicitly generated. Thus, the size of the games is only limited by the amount of search effort we wish to expend. Secondly, the critical rate parameter serves as a proxy for game difficulty. At one extreme, if $\gamma = 0$, then every child at every choice node has the same value as its parent — in effect, there are no "wrong" moves for either player, and planning becomes trivial. At the other extreme, if $\gamma = 1$, then every sub-optimal move at every choice node leads to a loss and the game becomes unforgiving. A single blunder at any stage of the game instantly hands the initiative to the opponent. Figure 2 gives examples of game trees generated with different settings of $\gamma$. For the sake of simplicity, we focus on trees with a uniform branching factor $b$ in this study.

### Critical Rates in Real Games

Before proceeding, we pause to validate our model by measuring the critical rates of positions in Chess (see Appendix E for similar data on Othello). We begin by first sampling a large set of positions that are $p$ plies deep into the game. These samples are gathered using two different methods:

- *Light playouts:* each side selects among the legal moves uniformly at random.

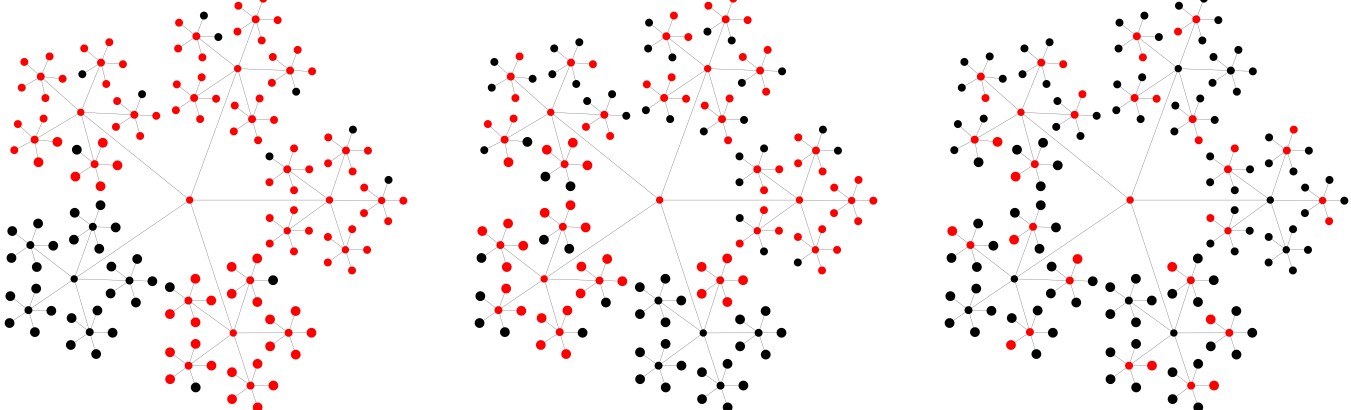

Figure 2: Effect of critical rate ($\gamma$) on game tree structure. Nodes in red correspond to $+1$ nodes, while nodes in black correspond to $-1$ nodes, with the root node in the center. The tree instances were generated with $\gamma = 0.1$, $\gamma = 0.5$, and $\gamma = 1.0$, from left to right.

- *Heavy playouts:* each side runs a 10-ply search using the Stockfish 13 Chess engine (Romstad et al. 2017) (freely available online under a GNU GPL v3.0 license) and then selects among the top-3 moves uniformly at random.

We approximate $v(s)$ for these sampled states using deep Minimax searches. Specifically, we use $v(s) \approx \text{sgn}\,(\tilde{v}_d(s))$, where $\tilde{v}_d(s)$ denotes the result of a $d$-ply Stockfish search. To compute the empirical critical rate $\tilde{\gamma}(s)$ for a particular choice node $s$, we begin by computing $\tilde{v}_{20}(s)$ and $\tilde{v}_{19}(s')$ for all the children $s'$ of $s$ and then calculate:

$$\tilde{\gamma}(s) = \frac{1}{b-1} \sum_{s'} \mathbb{1}[\text{sgn}\,(\tilde{v}_{19}(s')) \neq \text{sgn}\,(\tilde{v}_{20}(s))]$$

Admittedly, using the outcome of a deep search as a stand-in for the true game theoretic value of a state is not ideal. However, strong Chess engines are routinely used in this manner as analysis tools by humans, and we thus believe this to be a reasonable approach. Figure 3 presents histograms of $\tilde{\gamma}$ data collected for $p = 10$ and $p = 36$, using both light and heavy playouts. Each histogram aggregates data over $\sim 20,000$ positions.

We note that about 40–50% of the positions sampled have $\tilde{\gamma}$ values higher than 0.9, which is consistent with Chess's reputation for being a highly tactical game. We also see that the $\tilde{\gamma}$ values collected for Chess form a distribution that is non-stationary with respect to game progression, unlike in our proposed game tree model where $\gamma$ is fixed to be a constant. Nonetheless, we believe that this simplification in our modeling is reasonable: at deeper plies, the distribution of $\tilde{\gamma}$ becomes strikingly bimodal, with most of the mass accumulating in the ranges $[0.0, 0.1]$ and $[0.9, 1.0]$. This clustering means that one could partition Chess game tree into two very different kinds of subgames (with high and low $\gamma$), within each of which the critical rate remains within a narrow range.

## Heuristic Design

Before we can run UCT search experiments on critical win-loss games, we need to resolve one more issue: *how should*

*UCT estimate the utility of non-terminal nodes?* One popular approach to constructing artificial heuristics is the additive noise model — the heuristic estimate $h(s)$ for a node $s$ is computed as $h(s) = v(s) + \epsilon$, where $\epsilon$ is a random variable drawn from a standard distribution, like a Gaussian (Luštrek, Gams, and Bratko 2005; Ramanujan, Sabharwal, and Selman 2011). However, as we will see, static evaluations of positions in real games often follow complex distributions.

To better understand the behavior of heuristic functions in real games, we once again turn to Chess and the Stockfish engine. We sample $\sim 100,000$ positions each using light and heavy playouts for $p = 10$. As before, we use $\text{sgn}\,(\tilde{v}_{20}(s))$ as a proxy for $v(s)$ for each sampled state $s$. We also compute $\tilde{v}_0(s)$ for each $s$, which we normalize to the range $[0, 1]$ — this is the static evaluation of each $s$ without any lookahead. Figure 4 presents histograms of $\tilde{v}_0(s)$, broken out by $v(s)$. A clearer separation between the orange and blue histograms (i.e., between the evaluations of $+1$ and $-1$ nodes) indicates that the heuristic is better at telling apart winning positions from losing ones. Indeed, the ideal heuristic would score every $+1$ position higher than every $-1$ position, thus ordering them perfectly. We see in Figure 4 that such clear sorting does not arise in practice in Chess, particularly for positions that are encountered with strong play. Moreover, the valuations assigned to positions do not follow a Gaussian distribution, and attempts to model them as such are likely too simplistic. However, these histograms also suggest an empirical method for generating heuristic valuations of nodes — we can treat the histograms as probability density functions and sample from them. For example, to generate a heuristic estimate for a $-1$ node $s$ in our synthetic game, we can draw $h(s) \in [0, 1]$ according to the distribution described by one of the blue histograms in Figure 4. Of course, given the sensitivity of the shape of these histograms to the sampling parameters, it is natural to wonder *which* histogram should be used. Rather than make an arbitrary choice, we run experiments using a diverse set of such histogram-based heuristics, generated from different choices of $p$, different playout sampling strategies, and

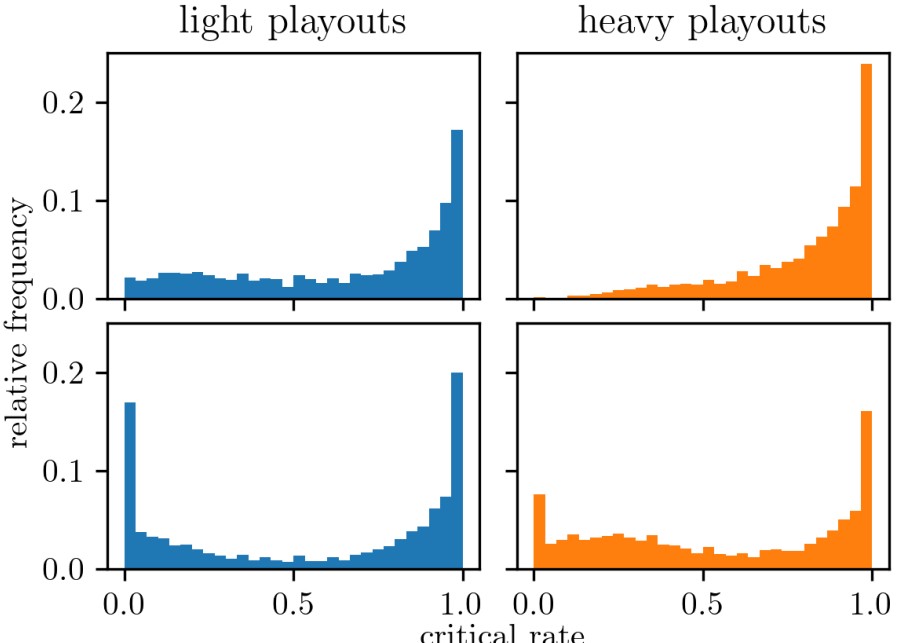

Figure 3: Histograms of empirical critical rates ($\tilde{\gamma}$) for Chess positions sampled $p = 10$ (top row) and $p = 36$ (bottom row) plies deep into the game. We sample the positions using both light playouts (left column) and heavy playouts (right column).

different game domains.

Additionally, we note that one can also use random playouts as heuristic evaluations, like in the original formulation of UCT. One advantage of our critical win-loss game tree model is that we can analytically characterize the density of $+1$ and $-1$ nodes at a depth $d$ from the root node, given a critical rate $\gamma$ and branching factor $b$. Specifically, for a tree rooted at a maximizing choice node, the density of $+1$ nodes at depths $2d$ and $2d + 1$ (denoted as $f_{2d}$ and $f_{2d+1}$ respectively) are given by:

$$f_{2d} = k^{2d} + \frac{1 - k^{2d+2}}{1 + k} \tag{1}$$

$$f_{2d+1} = f_{2d} \cdot k \tag{2}$$

where $k = 1 - \gamma + \gamma/b$. We refer the reader to Appendix B for the relevant derivations. Access to these expressions means that we can cheaply simulate random playouts of depth $2d$: the outcome of a single playout ($\ell_1$) corresponds to sampling from the set $\{+1, -1\}$ with probabilities $f_{2d}$ and $1 - f_{2d}$ respectively. For lower variance estimates, we can use the mean of this distribution instead, which would correspond to averaging the outcomes of a large number of playouts ($\ell_\infty$). In our experiments, we explore the efficacy of the heuristics $\ell_1$ and $\ell_\infty$ as well.

## 4  Results

**Theoretical Analysis**

We begin with our main theoretical result and provide a sketch of the proof.

**Theorem 1.** *In a critical win-loss game with $\gamma = 1.0$, UCT with a search budget of $N$ nodes will exhibit lookahead pathology for choices of the exploration parameter $c \geq \sqrt{\frac{N^3}{2 \log N}}$, even with access to a perfect heuristic.*

The key observation underpinning the result is that the densities of $+1$ nodes in both the optimal and sub-optimal subtrees rooted at a choice node (given by equations (1) and (2)) begin to approach the same value for large enough depths. This in turn suggests a way to lead UCT astray, namely to force UCT to over-explore so that it builds a search tree in a breadth-first manner. In such a scenario, the converging $+1$ node densities in the different subtrees, together with the averaging back-up mechanism in the algorithm, leaves UCT unable to tell apart the utilities of its different action choices. Notably, this happens even though we provide perfect node evaluations to UCT (i.e., the true minimax value of each node) — the error arises purely due to the structural properties of the underlying game tree. All that remains to be done is to characterize the conditions under which this behavior can be induced, which is presented in detail in Appendix C. Our experimental results suggest that the bound in Theorem 1 can likely be tightened, since in practice, we often encounter pathology at much lower values for $c$ than expected. We further find that the pathology persists even when we relax the assumption that $\gamma = 1.0$, as described in the following sections.

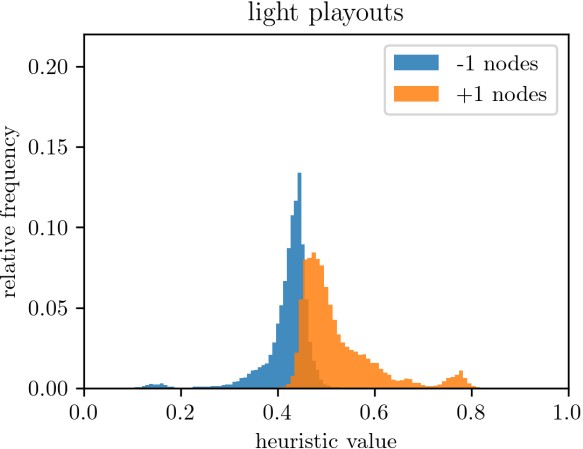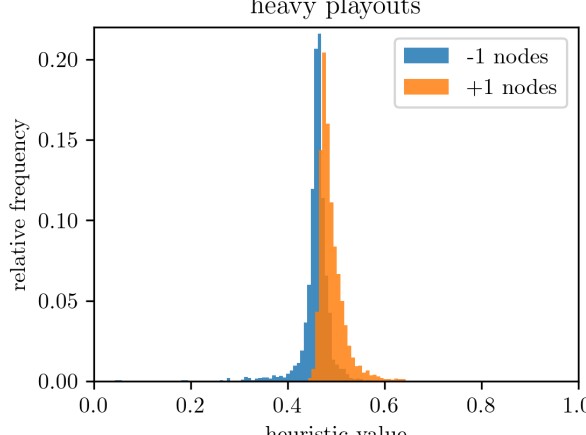

Figure 4: Distribution of Stockfish 13 static evaluations of $+1$ and $-1$ positions sampled $p = 10$ plies deep into Chess. The positions are sampled using both light playouts (left) and heavy playouts (right).

## Experimental Setup

We now describe our experimental methodology for investigating pathology in UCT. Without loss of generality, we focus on games that are rooted at maximizing choice nodes (i.e., root node has a value of $+1$). We set the maximum game tree depth at 50, which ensures that relatively few terminal nodes are encountered within the search horizon (other depths are explored in the supplementary material, see Appendix F). We present results from a 4-factor experimental design: 2 choices of critical rate ($\gamma$) × 3 choices of branching factor ($b$) × 2 choices of heuristic models × 5 choices of the UCT exploration constant ($c$). A larger set of results, exploring a wider range of these parameter settings, is presented in Appendices F–H. For each chosen parameterization, we generate 500 synthetic games using our critical win-loss tree model. We run UCT with different computational budgets on each of these trees, as measured by the number of search iterations (i.e., the size of the UCT search tree). We define the *decision accuracy* (denoted as $\delta_i$) to be the number of times that UCT chose the correct action at the root node, when run for $i$ iterations, averaged across the 500 members of each tree family sharing the same parameter settings. Our primary performance metric is the *pathology index* $\mathcal{P}_j$ defined as:

$$\mathcal{P}_j = \frac{\delta_j}{\delta_{10}}$$

where $j \in \{10, 10^2, 10^3, 10^4, 10^5\}$. Values of $\mathcal{P}_j < 1$ indicate that additional search effort leads to worse outcomes (i.e., pathological behavior), while $\mathcal{P}_j > 1$ indicates that search is generally beneficial. We ran our experiments on an internal cluster of Intel Xeon Gold 5128 3.0GHz CPUs with 512G of RAM. We estimate that replicating the full set of results presented in this paper, with 96 jobs running in parallel, would take about three weeks of compute time on a similar system. The code for reproducing our experiments is hosted at https://github.com/redacted-for-double-blind-review.

## Discussion

Figure 5 presents our main results. Our chief finding is that the choice of $\gamma$ is the biggest determinant of pathological behavior in UCT. For games generated with $\gamma = 1$, we find that UCT exhibits lookahead pathology *regardless* of all other parameters — the exploration constant, the branching factor, or how the heuristic is constructed. Appendices F and H confirm the robustness of this result using additional data collected for other game tree depths and for heuristics constructed using data collected from Othello. For smaller values of $\gamma$, the effect is not as strong and other factors begin to play a role. For example, with $\gamma = 0.9$, pathological behavior is most apparent at higher branching factors and with more uniform exploration strategies (i.e., higher settings of $c$), consistent with Theorem 1. For $\gamma = 0.5$, pathology is almost completely absent, regardless of other parameters (see Appendix G).

Poor performance from UCT in a synthetic domain, particularly when it is forced to over-explore, may seem unsurprising at first glance — so we will pause here to address these concerns, clearly elucidate the significance and novelty of these results, and to contextualize them better. Firstly, we note that lookahead pathology is distinct from poor planning performance. Practitioners routinely encounter domains where MCTS-style approaches simply fail to produce good results (for example, see (Ramanujan, Sabharwal, and Selman 2010a,b)), regardless of the size of the computational budget, but our work demonstrates a subtly different phenomenon. Namely, there are situations where UCT will initially make good decisions (when it is allowed to build a tree of size, say, 100 nodes), but that its performance will degrade as it is afforded more "thinking" time (on trees of size 10k nodes). Moreover, we have done so in a two-player setting, where winning strategies for a player need to be robust to any counter-move by the opponent. In practice, this means that there are an exponential number of winning leaf nodes for both players. Thus, demonstrating degenerate behavior

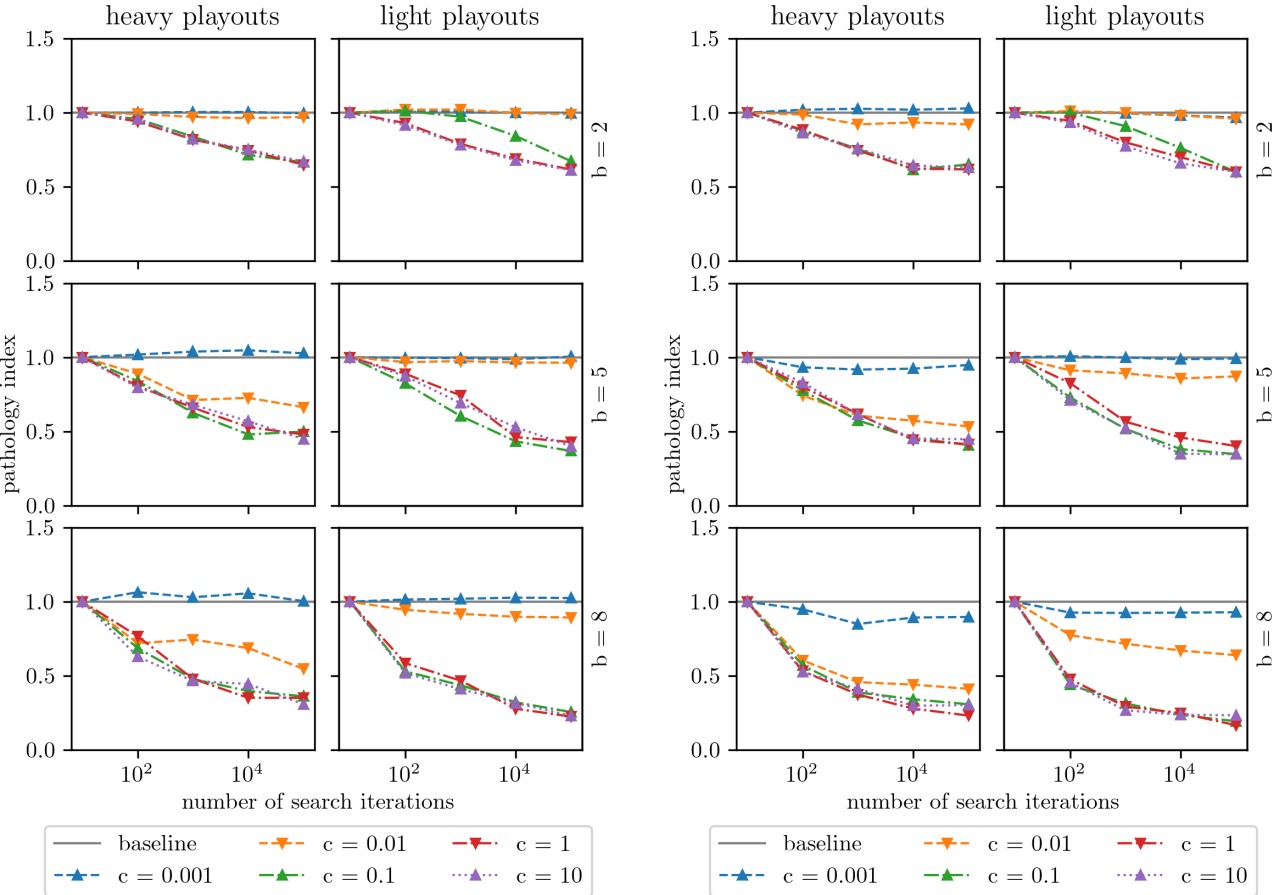

Figure 5: Measuring pathological behavior in UCT on critical win-loss games of depth 50 with $\gamma = 0.9$ (left) and $\gamma = 1$ (right). The heuristic to guide UCT is constructed from histograms of Stockfish evaluations of positions sampled at depth 10, using both light and heavy playouts. Each colored line corresponds to an instantiation of UCT with a different exploration constant. Note that the $x$-axis is plotted on a log-scale.

using constructions such as those of Coquelin and Munos, where the optimal leaf node is strategically hidden in the midst of an exponential number of suboptimal nodes, are not possible (Coquelin and Munos 2007). In fact, our model is symmetric; we use the same critical rate $\gamma$ for both players so that the game is equally (un)forgiving for both players, without stacking the deck against one player or the other. Seeing pathology arise under these conditions is thus more unexpected. We also note that more uniform exploration has been proposed in the past as an antidote to undesirable behavior in UCT (Coquelin and Munos 2007). Our results indicate that over-exploration may create new problems of its own.

Finally, for the sake of completeness, we also evaluate the performance of Minimax search with alpha-beta pruning on our critical win-loss games. These results are presented in Appendix J. We find that Minimax is similarly susceptible to lookahead pathology in our setting, particularly in games

where $\gamma \geq 0.7$. Games with high critical rates correspond to those with a high clustering factor $f$ (Sadikov, Bratko, and Kononenko 2005), and thus, low *local similarity* — in such games, there is a lower degree of correlation among the true utilities of nodes that are near each other in the game tree, which has been identified as a key driver of pathology (Nau et al. 2010). Our results are thus consistent with the findings of Nau et al. (Nau et al. 2010). One point of departure from their findings, however, is that pathology arises in our experiments even though we use a highly granular heuristic. In their work, Nau et al. create heuristic estimates for node utilities via an additive Gaussian noise model (Nau et al. 2010), whereas ours are derived from binning Stockfish's evaluation function and treating that as a distribution. This suggests that the manner in which heuristics are modeled may contribute to pathology, in addition to their resolution.

## Broader Impacts

This paper highlights a counter-intuitive failure mode for MCTS that deserves broader appreciation and recognition from researchers and practitioners. The fact that UCT has the potential to make worse decisions when given additional compute time means that the algorithm needs to be used with greater care. We recommend that users generate scaling plots such as those shown in Figure 5 to better understand whether UCT is well-behaved in their particular application domain, before wider deployment.

## 5 Conclusions

In this paper, we explored the question of whether MCTS algorithms like UCT could exhibit lookahead pathology — an issue hitherto overlooked in the literature. Due to the shortcomings of existing synthetic game tree models, we introduced our own novel generative model for extensive-form games. We used these critical win-loss games as a vehicle for exploring search pathology in UCT and found it to be particularly pronounced in high critical rate regimes. Important avenues for follow-up work include generalizing the theoretical results presented in this paper to games where $\gamma \neq 1$ and deriving tighter bounds for the exploration parameter $c$, as well as investigating whether such pathologies emerge in real-world domains.

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
