# OpenReview forum: "Lookahead Pathology in Monte-Carlo Tree Search"
_icaps-conference.org/ICAPS/2024/Conference — ICAPS 2024_

### Official Review · Reviewer_dx71 · 2024-01-03

**Significance And Importance:** 2
**Soundness:** 4
**Novelty:** 3
**Clarity:** 4
**Overall Evaluation:** 2
**Confidence:** 4

**Weaknesses:**

1: Minor weaknesses that are easily fixable.

**Contributions Of The Paper:**

-----------------
post-rebuttal:
Sorry to be unclear - by "hypothesis" I mean a (more or less speculative) explanation for the phenomenon that you have uncovered, what causes it, why it occurs (which could also include speculation on how it might be fixed or avoided). A more thorough discussion of these issues would be greatly appreciated by the reader. Additional experiments with min/max backups would seem to me to begin to test (what I understand to be) your hypothesis, so it would be great if they could be included (even if only in the appendix, but you currently have plenty of space in the paper).

Including your anecdotal results with previously-proposed tree models would definitely strengthen the paper.
-----------------
This paper presents a new synthetic game tree model and demonstrates that both minimax and UCT exhibit pathological behavior (worse decision-making as the amount of lookahead search increases) for a range of parameter settings.

**Ethical Considerations:**

(1) Not Applicable: The paper does not have any ethical considerations to address

**Nomination For Best Paper:**

No

**Questions For Authors:**

Would substituting min/max solve the problem?

Are there tests of your hypothesis that would be feasible to add to the paper?

What justifies using a new tree model rather than providing results on the old ones?  (I see this as a separate question from why is the new tree model better, which the paper already argues nicely.)

**Reproducibility:**

4: Authors promise to release code and domains (whichever apply).

**Strengths Of The Paper:**

Given the popularity of applying MCTS to games, demonstrating lookahead pathology is an important contribution.

The new synthetic game tree model is nice: it can be generated lazily from the root downward and the "difficulty" (number of children with the same win/loss value as the their parent) can be adjusted by a parameter.

The writing is clear and flows well.

If one considers the appendices, the empirical results are impressively comprehensive.

**Weaknesses Of The Paper:**

The submission devotes only a few lines to explaining why UCT exhibits pathology on the proposed trees.  I wished for more detail here, since this appears to be the core of the paper and there is plenty of room before hitting the page limit.  Furthermore, the hypothesis given is, as far as I can tell, not tested in any of the experiments.  The crux of the problem appears to be the fact that MCTS averages returns within each subtree of the root rather than, for example, taking mins and maxs to approximate the minimax value.  Would substituting min/max solve the problem?  On a related note, there is no discussion of how this flaw in adversarial applications relates to the guarantee of UCT to converge to the optimal action choice in an MDP.

The paper does not demonstrate that any of the previously-proposed tree models used to demonstrate pathology in minimax are insufficient to show pathology in UCT.  This would clarify how the new results relate to existing work, and how UCT compares to minimax.  (It would also be neat to see examples where minimax is not pathological but UCT is, but perhaps that's another paper.)

The relation of the new parameter gamma to the previously-proposed problem feature "local similarity" (Nau et al, 2010) is not explained either empirically or theoretically.  In essence, this paper is proposing an alternative set of conditions for pathology which will then need to be reconciled with the previous explanation of pathology for minimax.  This fragmentation is unfortunate and it is not clear that it is necessary.


little things:

1.1 (page 1 line 1) my understanding is that UCT was proposed for MDPs and its use for adversarial search is a bastardization.  is the use of the adjective "adversarial" really justified here?

in multiple places, the paper intends to use the authors of a specific reference as a subject.  a nice way of doing this is to say "Jones (2017) reported ...".  the current manuscript uses "Jones reported ... (Jones 2017)" which seems redundant and takes more space.  eg 1.57

1.71 my understanding is that Nau et al (2010) showed that pathology arises in real games such as mancala and chess, not just in synthetic trees.  perhaps this is important enough to mention here?

2.107 I assume \overbar{Q} is meant, instead of Q?

2.113 citation typo

2.115 P-games have not been introduced yet

2.128 would it be worth mentioning that lookahead pathology has also been found in real-time search (eg, the work of Bulitko)?

3.185 "determine the conditions" - that's a tall order!  perhaps you want to limit the reader's expectations by something qualified like "determine some sufficient conditions"?

3.225 I understand that this is a subjective point on which the authors should have the final say, but I wanted to mention that I am not entirely comfortable with the term "critical rate".  it only indirectly controls the rate at which critical moves are generated (and the connection is never mentioned, let alone explored, in the paper at all).  a more concrete term would be something like "value flip probability".

3.236 I feel that this definition is potentially ambiguous: is there one random draw to decide the value of all non-optimal children, or a separate one for each?

4.285-290 it is not clear why this implies reasonableness

4.315 pedantry: does an optimal heuristic really need to put all +1 positions above every -1 position?  perhaps there is a proper subset of -1 states against which a +1 position is actually compared (directly or indirectly)?

4.320 "likely too simplistic" - why?  please clarify/explain your argument here.

it would be nice if the columns and rows in the figure matrices were labeled (currently info is only in caption)

5. 334 personally I find 1 - \gamma (1 + \frac{1}{b}) more intuitive, but this is subjective

5.361 more detail here would be great.  is this a UCT-specific problem (would you expect UCT to fare worse than minimax)?  does this hypothesis imply that UCT will do worse on deeper states than shallower ones (when number of iterations is held constant)?

how will the appendices be handled in the final version?  reference to an arXiv version?

6.399 not clear what "uniform" means here

6.404 how can you tell that UCT is being forced to over-explore?  why would this hurt the algorithm more when more iterations are used?

6.454-456 this was not clear to me - why does this suggest that?

---

> ### Author Rebuttal · Authors · 2024-01-28
>
> We would like to thank the reviewer for their thoughtful comments and very thorough feedback. We address their questions below.
>
> **The submission devotes only a few lines to explaining why UCT exhibits pathology on the proposed trees. [...] Would substituting min/max solve the problem? ...**
>
> We would like to thank the reviewer for this interesting question! In the theoretical analysis we perform in the paper, we consider a regime where the search agent has access to a perfect heuristic: clearly, in this scenario, min/max backups would necessarily yield the optimal outcomes (at least when a given level of the game tree has been completely expanded). The answer is less clear when one moves away from perfect heuristics, and this would require empirical investigation. One piece of related (empirical) work in this area is that of Coulom (2006), who found that a hybrid back-up strategy worked best in the early Go-playing engine CrazyStone – his engine used averaging backups for nodes with low visit counts, but minimaxing at nodes with high visit counts.
>
> R. Coulom, “Efficient Selectivity and Backup Operators in Monte-Carlo Tree Search”, 5th International Conference on Computer and Games, 2006.
>
> **Are there tests of your hypothesis that would be feasible to add to the paper?**
>
> We’re not completely sure we understand what the reviewer is referring to by the “hypothesis”. If they are referring to the question of whether the averaging backups in UCT are the main culprit, and are recommending some additional experiments with min-max backups, to examine whether pathology vanishes in that setting – we would be happy to run these experiments and add them to the paper or the appendices, as a supplementary discussion, if the paper is accepted.
>
> **What justifies using a new tree model rather than providing results on the old ones?...**
>
> We started this line of research by actually considering some of the existing synthetic game tree models, but we quickly ran into their various limitations which prevented us from running the experiments we wanted to run. For example: with the original P-games proposed by Pearl and Nau, we did not observe pathological behavior from UCT with shallow binary trees (depth of ~20–30). We hypothesized that things might be different with larger games, but needing to hold the entire tree in memory limited what we could explore (larger depths or branching factors). Hence why we resorted to proposing and studying our own models.

---

### Official Review · Reviewer_oPUt · 2024-01-17

**Significance And Importance:** 2
**Soundness:** 3
**Novelty:** 3
**Clarity:** 3
**Overall Evaluation:** 1
**Confidence:** 3

**Weaknesses:**

0: Minor weaknesses requiring some work to be addressed for the paper to be accepted.

**Contributions Of The Paper:**

The authors discuss pathological behaviors of MCTS, namely UCT, which is a standard algorithm in the game/planning research community.  They prove a theorem which shows when a pathology occurs even with a perfect heuristic.  They also introduce a new synthetic game tree to analyze search behaviors and empirically show UCT may suffer from pathologies with heuristic functions whose value distributions are similar to chess (and Othello shown in the appendix).

**Ethical Considerations:**

(1) Not Applicable: The paper does not have any ethical considerations to address

**Nomination For Best Paper:**

No

**Questions For Authors:**

Could you have any response to the first and second weaknesses I pointed out?

**Reproducibility:**

4: Authors promise to release code and domains (whichever apply).

**Strengths Of The Paper:**

This is a well-written paper and includes a lot of interesting insights, including an introduction of a new synthetic game, and important empirical analysis with an attempt to model chess and Othello in the form of synthetic games.

**Weaknesses Of The Paper:**

- It is unclear how Theorem 1 is correlated to the experiments where parameter c is set to small. In my understanding, in Theorem 1, a large value of c is necessary for a pathological behavior. For example, for N=50, c needs to be set to c>126. Such a large parameter leads UCT to merely selecting a move with the highest node visit, because the exploitation term for accumulating the rewards becomes much smaller than the value for the exploration term. Therefore, Theorem 1 just indicates a natural outcome of the pathology caused by ignoring the reward and selecting a move with the highest visit count.

- I wonder why the authors do not show the distribution for the game of Go. Both chess and Othello are the games in which UCT has had a difficulty in outperforming alpha-beta (with a recent exception of AlphaZero in chess). In contrast, Go is a game where alpha-beta has never worked efficiently than UCT. Analysis with a Go-like distribution would be important in the community even if a pathology is not observed.

This is a very minor point but parallel UCT in the following paper uses a synthetic game for performance measurement. Although they use a different type of a synthetic game and for a different purpose, I think it must be cited as related work, considering they use the synthetic game.

Kazuki Yoshizoe, Akihiro Kishimoto, Tomoyuki Kaneko, Haruhiro Yoshimoto, Yutaka Ishikawa: Scalable Distributed Monte-Carlo Tree Search. SOCS 2011: 180-187

---

> ### Author Rebuttal · Authors · 2024-01-28
>
> We would like to thank the reviewer for their thoughtful comments and feedback. We address their questions below.
>
> **It is unclear how Theorem 1 is correlated to the experiments where parameter c is set to small...**
>
> We acknowledge the reviewer’s concern, but we note that Theorem 1 and the experimental results complement each other. We carry out the theoretical analysis in a regime where this is feasible – when using a perfect heuristic and when $\gamma=1.0$, to establish bounds on $c$.  This establishes the theoretical possibility of pathological behavior in UCT. The experiments relax these assumptions and demonstrate that, in fact, pathology arises in a broader range of parameterizations, including regimes that are challenging to analyze theoretically.
>
> **I wonder why the authors do not show the distribution for the game of Go...**
>
> Like the reviewer, we too were interested in Go as our second domain of interest. However, it quickly became apparent after some initial experiments that collecting the volume of data that was needed to conduct this study was infeasible using engines like KataGo. Analyzing even a single Go position was often orders of magnitude slower than a comparable analysis using Stockfish or Edax. And when one considers the total experimental cost of our work, we estimate that replicating our full suite of experiments would require about 3 weeks of compute time. Doing this for Go would have thus resulted in months of compute time, given the resources available to us. Our compromise was thus to study Othello, a game with Go-like characteristics (two-player game, with similar concepts of territory capture and control), for which much faster game engines are available.
>
> **This is a very minor point but parallel UCT in the following paper uses a synthetic game for performance measurement...**
>
> We thank the reviewer for this pointer. This paper uses the N-game model we discuss in our “Synthetic Game Tree Models” subsection (though Yoshizoe et al. refer to it as a P-game), so this family of models is indeed part of our discussion. However, for the sake of completeness, we will include this reference in the final version of our paper, if accepted.

---

### Official Review · Reviewer_qTeB · 2024-01-22

**Significance And Importance:** 2
**Soundness:** 2
**Novelty:** 2
**Clarity:** 2
**Confidence:** 2

**Weaknesses:**

-1: Major weaknesses requiring significant work to be addressed for the paper to be accepted.

**Contributions Of The Paper:**

- A model of games that is parameterised (\gamma) in difficulty, and games in which the tree depth can be arbitrarily large.
- Partial theoretical analysis

**Ethical Considerations:**

(1) Not Applicable: The paper does not have any ethical considerations to address

**Nomination For Best Paper:**

No

**Overall Evaluation:**

-1: (weak reject)

**Questions For Authors:**

- How can one formally detect a pathology? Moreover, formally/mathematically define a pathology.

- There is an inconsistency in the explanation in section "Game Tree Model". You first say that choice nodes must have at least one child with the same minimax value. In the following paragraph, you emphasize that "exactly one" child of a choice node must share the minimax value. Please clarify or correct me if I am wrong. This is an important piece of explanation as it is about one of the main contributions.

Small refinements:
- Adjust figure one, it is currently stacked instead of side-by-side.
- Consider Figure 2 nodes in red to a more contrasting colour, or a lighter red, to make the visualisation easier (personal opinion). It would become easier to visualise if printed as well.

**Reproducibility:**

4: Authors promise to release code and domains (whichever apply).

**Strengths Of The Paper:**

- The introduction of a game model to analyse lookahead pathology contributes to the literature.
- A theoretical analysis when critical rate = 1.

**Weaknesses Of The Paper:**

- Although a theoretical analysis is given for games with critical rate = 1 is a good starting point, it would be interesting to see if the proposed game model is able to bring us good theoretical results on values != 1.
- The main aspect that should be analysed in the paper, which is lookahead pathology, is not mathematically defined.
- There is a lot of focus on the introduction of the Critical Win-Loss games and the critical rate parameter . The game model is an important contribution. However, I feel a lot is missing on explaining and analysing what the paper title actually attracts the reader for, which is lookahead pathology.

---

> ### Author Rebuttal · Authors · 2024-01-28
>
> We would like to thank the reviewer for their thoughtful comments and feedback. We address their questions below.
>
> **How can one formally detect a pathology? Moreover, formally/mathematically define a pathology.**
>
> Lookahead pathology was defined by Nau (1979, 1980) as when the decision accuracy of search approaches that of random decision-making as the search depth increases to infinity. We adapt this definition to the MCTS setting (essentially: replace search depth with the number of iterations), and quantify it formally using the pathology index $\mathscr{P}_j$ defined in the “Experimental Setup” subsection under Section 4. We will draw more attention to this definition by moving it earlier in the paper, if our submission is accepted.
>
> The question of _detecting_ pathology is a little trickier, since to do it an error-proof way, one needs access to the ground truth (i.e., the correct decision to make at the root node) – but in such a situation, there is no need for search! One could devise other heuristic methods to detect pathology, but we believe that such exploration is beyond the scope of this paper which is primarily focused on illustrating this little studied phenomenon in the context of MCTS.
>
> **There is an inconsistency in the explanation in section "Game Tree Model"...**
>
> We agree with the reviewer that the wording in these two sentences is confusing. At least one child of a choice node must have the same minimax value as its parent. This is equivalent to saying that one child of a choice node _must_ have the same minimax value as its parent, with the values of the remaining children being unconstrained. We will drop the word “exactly” in the second sentence, which muddies the waters.
>
> **Small refinements:**
>
> We thank the reviewer for this feedback – it will improve the visual presentation in the paper, and we will make these updates in the final version, if the submission is accepted.

---

### Meta-Review · Area_Chair_g8LH · 2024-02-06

**Recommendation:** Accept (Poster)
**Confidence:** 4

**Metareview:**

This paper studied so-called "lookahead pathology" of UCT algorithm
where deeper search harms the performance.

The paper proposed a synthetic game tree model that can generate arbitrary large instances
and is parameterized by $\gamma$ for flexible configuration.
The authors first empirically analyzed Chess to see if its search space can be seen as an instance of such a synthetic game tree
with a particular $\gamma$.
They then proposed a theorem that when the exploration coefficient $c$ is large, pathology is possible.
#
Finally, by running
UCT guided by analytically available heuristics on the proposed synthetic model,
they confirmed the existence of pathology.

Although there were minor initial confusions,
the reviewers reached a consensus and generally found the paper valuable.
After the rebuttal the score changed +1/+1/+2, leaning toward acceptance.



Meta-reviewer comments:

Personally the characterization of $c$ feels odd from the multi-armed bandit standpoint;
Although it might be a common practice to tune $c$,
in UCB1's proof $c$ is the range of reward values,
which is 1 because rewards in the win/loss game is in [0,1],
i.e., theoretically the only correct value of $c$ in these games is $c=1$.

That being said, large $c$ does not invalidate the convergence characteristics of UCB1.
The asymptotic optimality (logarithmic regret) does not hold if the reward violates the range $[0,c]$,
but making $c$ larger does not cause such a violation.
As Reviewer oPUt pointed out, larger $c$ naturally causes issues even without Theorem 1,
but having the issue formalized is probably an important contribution.

For bounded-reward setting like games,
there are plenty of bandit algorithms other than UCB1,
although I do not know if they have been applied to adversarial games in a competitive setting.
For example, UCB-V selects actions according to $\mu + \sigma \sqrt{ (2\log T)/t } + (3c \log T)/t$
where $\sigma$ is a backed-up variance of Q while $\mu$ is a backed-up average of Q.
I wonder if the pathology still exists with those alternative bandits.

**Ethical Considerations:**

(1) Not Applicable: The paper does not have any ethical considerations to address